# Optogenetic silencing of hippocampal inputs to the retrosplenial cortex causes a prolonged disruption of spatial working memory

**Bárbara Pinto-Correia[1], Patrícia Caldeira-Bernardo[1], Miguel Remondes[1,2]\***

[1]Instituto de Medicina Molecular, Faculdade de Medicina, Universidade de Lisboa, Lisbon, Portugal; [2]Faculdade de Medicina Veterinária, Universidade Lusófona, Lisbon, Portugal

## eLife assessment

The authors report that optogenetic inhibition of hippocampal axon terminals in retrosplenial cortex impairs the performance of a delayed non-match to place task. Elucidating the role of hippocampal projections to the retrosplenial cortex in memory and decision-making behaviors is **important**. However, the strength of evidence for the paper's claims is **incomplete**.

**\*For correspondence:**
mremondes@medicina.ulisboa.pt

**Competing interest:** The authors declare that no competing interests exist.

**Abstract** Working memory allows us to keep information in memory for the time needed to perform a given task. Such fundamental cognitive ability relies on a neural circuit, including the retrosplenial cortex (RSC), connected to several cortical areas, functionally and anatomically, namely primary visual areas, and higher cognitive areas such as the cingulate, midcingulate, and subicular cortices. RSC bears intimate anatomical and functional connections with the hippocampus and has been implicated in integrating and translating spatial-temporal contextual information between ego- and allocentric reference frames to compute predictions about goals in goal-directed behaviors. The relative contribution of the hippocampus and retrosplenial cortex in working memory-guided behaviors remains unclear due to the lack of studies reversibly interfering with synapses connecting the two regions during such behaviors. We here used eArch3.0, a hyperpolarizing proton pump, to silence hippocampal axon terminals in RSC while animals perform a standard delayed non-match to place task. We found that such manipulation impairs memory retrieval, significantly decreasing performance and hastening decision-making. Furthermore, we found that such impairment outlasts light activation of the opsin, its effects being noticed up to three subsequent trials.

## Introduction

The ability to guide behavior according to knowledge stored previously in memory lies at the heart of adaptive behavior and is essential for survival. A crucial aspect of adaptive behavior is the capacity to make decisions based on the spatial structure of the surrounding context, something globally known as spatial cognition or spatial goal-directed behavior. Such cognitive function starts with the selection of relevant information from a flood of sensory stimuli and ends with its storage in cortico-hippocampal circuits (*Morris et al., 1982*; *Nadel, 1990*; *Scoville and Milner, 1957*) ultimately as a cognitive map manifested in a vast repertoire of neural activity, selective to multiple ego- (self-referenced) and allocentric (world-referenced) spatial variables, the seminal example of which is the existence of hippocampal place-selective neurons (*O'Keefe and Dostrovsky, 1971*). To allow

for the planning of the movements necessary to reach an intended goal, all the while ensuring that body actions are consistent with the context acted upon, a mechanism must exist that reinstates (retrieves) allocentric contextual representations, stored in the hippocampus (HIPP) as spatial memories, aligned to the animal's viewpoint, e.g., from an egocentric perspective (*Bicanski and Burgess, 2018*; *Carstensen et al., 2021*; *Gomez et al., 2009*; *Gomez et al., 2013*; *Michelmann et al., 2020*). In such a circuit, the retrosplenial cortex (RSC) emerges as a cornerstone, encoding contextual variables in both ego- and allocentric reference frames, representing future spatial goals, and effectively anchoring spatial memory-dependent decisions (*Alexander et al., 2020*; *Alexander and Nitz, 2015*; *Carstensen et al., 2021*; *Carstensen et al., 2021*; *Gomez et al., 2009*; *Gomez et al., 2013*; *Michelmann et al., 2020*; *Nitzan et al., 2020*).

Despite this remarkable work, we are yet to determine which neural pathways convey allocentric-referenced information stored in the HIPP, through hypothetical recall and egocentric referencing in RSC, onto the anterior midline cingulate (CG) and medial prefrontal cortical areas, responsible for planning motor behavior according to previous events, present needs, and motivation (*Akam et al., 2021*; *Brown and Braver, 2005*; *Cowen et al., 2012*; *Jin et al., 2009*; *Miller et al., 2019*; *Sober and Sabes, 2003*). We have recently found that HIPP axons target the divisions of the medial mesocortex: CG, midcingulate, and RSC, according to a gradient, in which RSC receives a combination of anatomically and functionally distinct inhibitory and excitatory functional synapses (*Ferreira-Fernandes et al., 2019*), whose manipulation results in specific behavioral deficits (*Yamawaki et al., 2019*). We hypothesize that these direct connections between hippocampal CA1 and RSC convey recently encoded and recalled memories to the relevant medial mesocortical regions.

To test this hypothesis, we used the delayed non-match to place (DNMP) task, a spatial working memory (WM) task that requires animals to retrieve the memory of a previous decision, and of the time it occurred, to disambiguate it from similar, previously rewarded, decisions (*Aggleton et al., 1986*; *Maguire, 2001*; *Shaw and Aggleton, 1995*; *Shaw et al., 2013*; *Wilson et al., 2013*). We then used optogenetics to reversibly interrupt synaptic transmission between HIPP and RSC neurons during the retrieval phase of the DNMP task. Optogenetics uses light-activated membrane-bound molecules with the capacity to sustain and reversibly change neuronal membrane potential through various molecular mechanisms, namely opening ion channels or pumping ions across the membrane (*Fenno et al., 2011*), to excite or silence groups of neurons with behavior-specific temporal accuracy. As far as it is known, optogenetics is the only molecular technique allowing for reversible perturbation of neural activity within the time constants of the cognitive functions at play during WM (*Yamamoto et al., 2014*).

To test the hypothesis that HIPP inputs onto RSC carry contextual information necessary for DNMP performance and are thus necessary for spatial WM, we used the optogenetic neural silencer eArch3.0 expressed in hippocampal neurons via local injection of an anterograde, static, viral vector (reviewed in *Saleeba et al., 2019*), AAV5-hSyn1-eArch3.0, under a pan-neuronal promoter, or its 'empty' counterpart with only YFP, in adult male Long-Evans rats previously trained in DNMP to 75% performance. We then ran multiple 28-trial DNMP sessions, in which we randomly interleaved two types of trials: no-illumination (NI) or illumination during the test, or retrieval, phase (TI), with green (525 nm) light delivered via two diagonal, intra-cerebral lambda fiber-optic implants, longitudinally staggered such as to deliver light to the full extent of RSC. In each animal, we measured three performance variables: individual trial (error/correct) and global performance (% correct trials), response latency to the choice point, and time spent at the choice point. We then used multivariable generalized linear mixed model (GLMM) using the above categorical and continuous variables to investigate the impact of silencing hippocampal synapses onto RSC neurons in spatial WM-dependent behavior.

If indeed the HIPP input onto RSC neurons anchors the decisions during 'test' runs to the memory of the 'sample' runs, we predict that by silencing HIPP inputs in RSC, we will prevent the transfer of information previously encoded in HIPP, and thus hinder the mechanism allowing RSC to inform the correct trajectory during the 'test' run, resulting in lower performance, specifically in eArch+ animals.

## Results

### Injection of AAV-5-hSyn1-eArch3.0 results in significant expression of a fluorescent tag in RSC hippocampal terminals

Besides abundant expression of both eArch and CTRL transgenes in hippocampal CA1, we found highly significant expression in the RSC layers as previously reported (*Figure 1A—figure supplement 1*; *Ferreira-Fernandes et al., 2019*). We could observe that HIPP axon terminals are distributed with high density in the superficial RSC layers II–III, as well as in deeper layer V, and sparsely on all other layers, forming beads-on-a-string structures, typical of axon terminals establishing synapses. On these RSC regions, we surgically implanted two fiber-optic stubs diagonally to target both hemispheres, at a relative reverse angle in distinct AP coordinates (*Figure 1B*, Methods).

### DNMP sessions in which no light was delivered resulted in no differences in behavioral performance between eArch+ and CTRL animals

Freely moving behavior is conditioned by the relations of body shape and movement, with surrounding context structure and dynamics. As such, to make sure all behavioral alterations observed during light delivery and opsin activation are due solely to this mechanism, it is imperative that pre-illumination DNMP testing is run under the same conditions as those used while we silence RSC-HIPP axon terminals, except for light delivery. The DNMP task includes a sample (encoding) phase, a delay (memory retention) period, and a choice (retrieval) phase, on a T-shaped maze apparatus. During the sample run, animals are rewarded for running the only available arm, with the alternative arm being blocked. This is followed by the delay period, and the 'test' (retrieval) run, during which both arms are made available, and animals are rewarded for choosing the arm previously blocked (i.e. 'not matching' the arm sampled previously).

We thus ran the DNMP protocol in two groups of animals expressing either AAV-5-hSyn1-eArch3.0 (eArch+) or an empty construct (CTRL) AAV-5-hSyn1-YFP, with fiber-optic stubs surgically implanted, and patch cords attached to the fiber-optic stubs, no light being delivered (*Figure 2A and B*). Under such conditions, we found no differences between eArch+ and CTRL animals in any of the three sessions run (*Figure 2C*, *Supplementary file 1A*, GLMM, p=0.212, Bonferroni post hoc test).

### eArch activation via green light illumination resulted in significantly lower global performance in eArch+, but not CTRL, animals

After the 'baseline' phase above, we started the 'illumination' phase protocol, in which 'test-illuminated' trials (TI), with light delivered through the fiber-optic implants in the test epoch of the DNMP protocol, were randomly interleaved with 'non-illuminated' trials (NI), where no light was delivered in the same ~25 min session (*Figure 2B*). In these sessions, we observed a significant difference between eArch+ and CTRL animals, with an overall lower performance in eArch+ animals (*Figure 3A*, *Supplementary file 1B*, GLMM, p=0.006, Bonferroni post hoc test). In addition, we found that the ratio of correct trials in the TI and NI trial groups (*Figure 3B*, *Supplementary file 1A and B*) was both significantly different from baseline levels (p<0.001), and from their CTRL counterparts (TI, p=0.02, NI, p=0.008), whose performance did not differ from baseline (p=0.252, p=0.184). The fact that more errors were committed in both TI and NI trials of an illuminated session, but only in eArch+ animals, being absent from CTRL animals, implies two things: first, the effect we observe is not the result of light alone acting upon neurons, an effect described earlier in striatal medium spiny (*Owen et al., 2019*) and CG neurons (*Stujenske et al., 2015*); second, the effects of eArch light activation of hippocampal inputs onto RSC neurons *outlasted illumination*, affecting subsequent NI trials. We were surprised by this result, as it indicates that the effects of eArch activation on either, or both, the neurophysiology of HIPP-RSC synapses and the ensuing circuit mechanisms involved in DNMP last longer than would be expected. To directly address this possibility, we analyzed the probability of animals committing errors after illuminated trials. We observed that, overall, illuminated trials were followed by a decreased probability of correct trials on eArch+, but not CTRL, animals (*Figure 3C*, *Supplementary file 1D*, p=0.015). This is consistent with the effects of eArch activation outlasting illumination by enough time such as to affect subsequent trials.

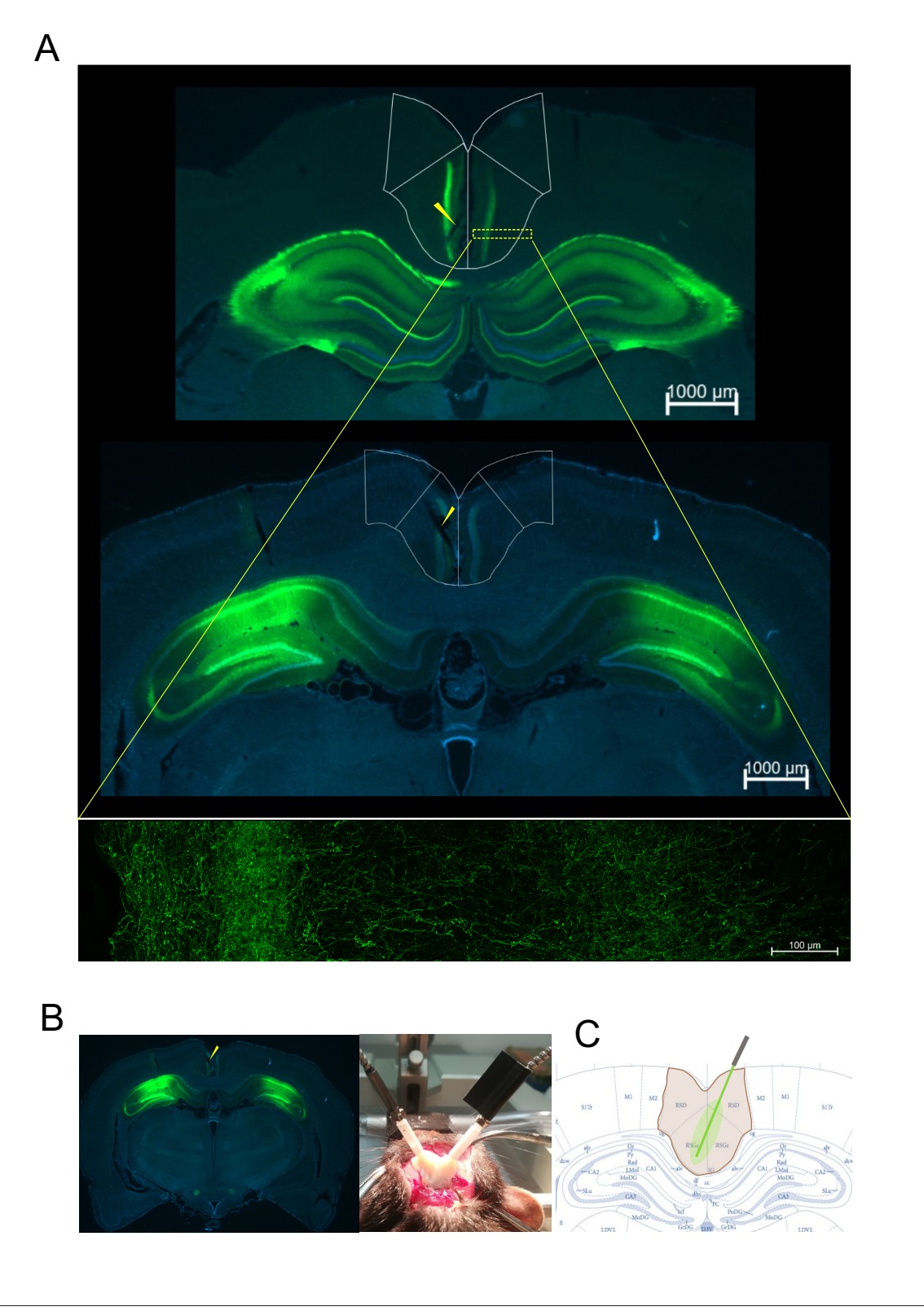

**Figure 1.** Viral vector expression and fiber-optic placement. (**A**) Anatomical distribution of eYFP fluorescence expression in distinct AP coordinates of the retrosplenial cortex (RSC) after injection in the hippocampus (HIPP). (**B**) Injection spots in the HIPP. (**C**) Fiber-optic placement with two fiber-optic stubs at AP-reversed 30° angles reaching the two hemispheres at two distinct AP levels of RSC (arrows point to fiber-optic tracks, please see ***Figure 1— figure supplement 1*** for low-magnification images of all animals and examples of high-magnification images).

*Figure 1 continued on next page*

*Figure 1 continued*

The online version of this article includes the following figure supplement(s) for figure 1:

**Figure supplement 1.** Viral vector expression and fiber-optic placement on all animals.

The effect of light delivery in subsequent non-illuminated trials in sessions where the two types of trials are interleaved within 1–2 min, as in the present study, has at least one precedent study in *Robinson et al., 2017*. In that study, trials in which a silencer opsin is activated by light are characterized by increased error probability and abnormal neural activity with decreased temporal selectivity and coding. Thus, both neural activity and behavior of interleaved non-illuminated trials are affected by the activation of the opsin in illuminated trials, leading authors to hypothesize that such persistent disruption of temporal coding in light-off trials is caused by synaptic plasticity events largely specific to the neural activity taking place in the (previous) epoch where light is delivered.

Our findings also imply that the notion of millisecond time-resolved neuron silencing using eArch is not applicable to synaptic transmission and, by consequence, to the ensuing behavioral effects we now investigate.

## Contrary to CTRL animals, eArch+ animals do not correct course following an incorrect decision in illuminated sessions

A fundamental principle of learning theory, and a common empirical finding in reward-based learning experiments, is that following an error in which reward is withheld, animals will correct their course of action. Conversely, following a correct decision, which is rewarded, animals maintain their course of action. This behavior is presumably informed by the association of the previous action with resulting outcome, which they store in memory for the next trial (*Mizumori and Jo, 2013*). Under this assumption, we predict that in eArch+, but not CTRL animals, illuminated trials will be followed by persistently defective HIPP-RSC synaptic function, presumably preventing subsequent decisions from anchoring themselves to the previous action-outcome association, even in the absence of light delivery, resulting in a higher probability of errors. To test such prediction, we analyzed the probability of an error trial following error, in illuminated sessions, and found that it is significantly higher in eArch+ animals than in CTRL animals, regardless of whether the present trial is itself illuminated (*Figure 4A*, *Supplementary file 1E*, p<0.001, Bonferroni post hoc test). Importantly, this difference is absent in baseline sessions, in which no illumination occurs (*Figure 4B*, *Supplementary file 1F*, p=0.172, Bonferroni post hoc test). This is consistent with our prediction and suggests that silencing HIPP terminals in RSC interrupts the flow of information from the previous trial, preventing subsequent error correction for which it is necessary.

## How long does the effect of eArch silencing last?

The silencing effects of light-mediated eArch activation on hippocampal axon terminals onto RSC neurons outlasted illumination, affecting subsequent trials and lowering spatial WM performance. We thus sought to determine how long would such an effect last. According to existing literature, once expressed in CA3-CA1 synaptic terminals, illumination of Arch with green-yellow light results in its activation into a powerful outward-moving proton pump, effectively hyperpolarizing neurons, independent of local membrane voltage (*Krol et al., 2019*; *Miao et al., 2015*). Besides this effect, eArch was recently found to reduce the amplitude and slope of fEPSP from right after the onset to over 2 min after illumination offset, via a pH increase that is independent of hyperpolarization, and to cause a sustained decrease in synaptic vesicle release (*El-Gaby et al., 2016*). Such sustained alkalinization, specific to synaptic boutons, lasts for several minutes post-activation of eArch and has been replicated on striatal-cortical synapses as well (*Mahn et al., 2016*). This previous data reporting the effects of eArch activation led us to further investigate, in our experimental conditions, how far in time would eArch behavioral effects last. We thus asked the following question: if one trial is illuminated, how many subsequent trials will be affected? To answer this question, we computed the probability of a correct trial on trial t+1, t+2, t+3, and t+4, following every illuminated trial. Our previous data shows that, overall, non-illuminated trials suffer the effects of illumination, resulting in more errors. However, by selecting the second previous and third previous trial as a trigger for our analysis, we will be biasing our analysis to include a few sequences of one or two interleaved non-illuminated trials, thus possibly

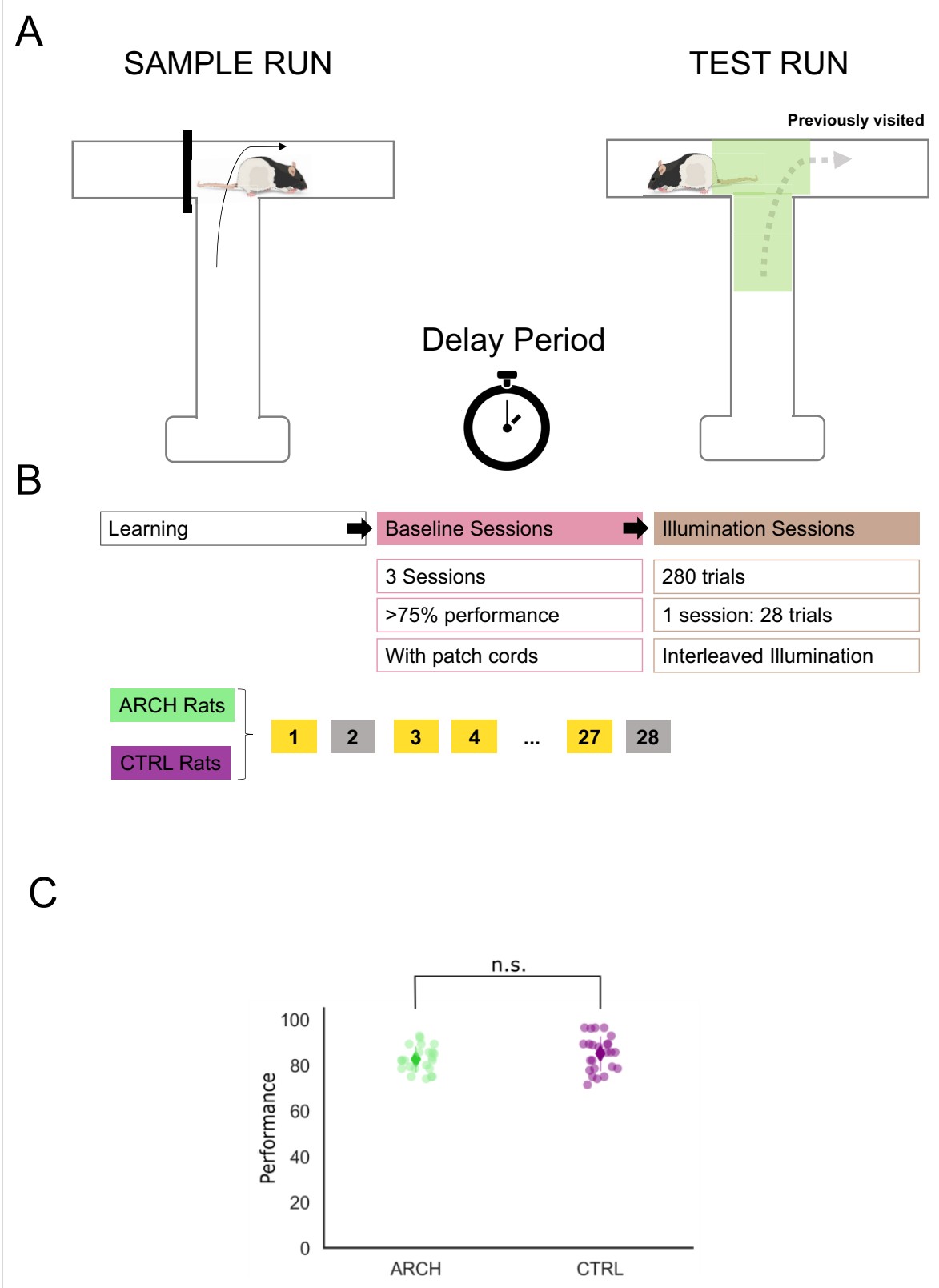

**Figure 2.** Delayed non-match to trajectory protocol and baseline performance. (**A**) Delayed non-match to place (DNMP) protocol. Rats explore the unblocked arm in a sample run, then wait in the starting area for a 15 s delay. Once back on the track, with both arms unblocked, rats receive a reward if they choose the arm opposite the one explored in the sample run. (**B**) Training schedule, after reaching 75% performance, 3 baseline (no light delivered) sessions are followed by 10 sessions with illuminated trials interleaved with non-illuminated ones. (**C**) No differences in performance were observed in the baseline sessions (generalized linear mixed model [GLMM], p=0.212, Bonferroni post hoc test).

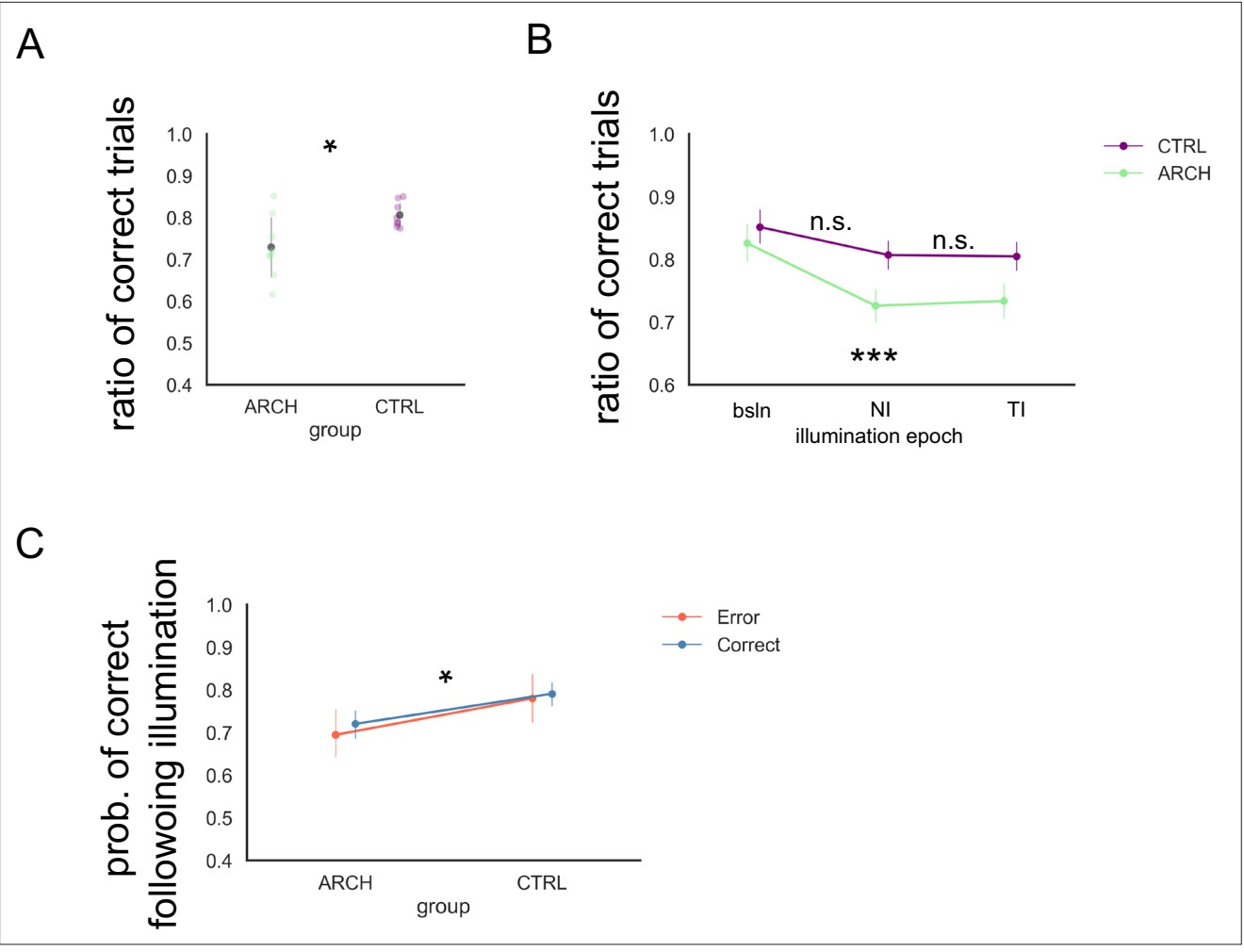

**Figure 3.** Optogenetic silencing of hippocampus (HIPP) terminals in retrosplenial cortex (RSC) causes a significant and prolonged decrease in performance. (**A**) Illumination of RSC during delayed non-match to place (DNMP) leads to a significantly lower performance in eArch+, compared to empty-vector CTRL animals (CTRL: 0.81±0.03, ARCH: 0.73±0.08, generalized linear mixed model [GLMM], p=0.006, Bonferroni post hoc test). (**B**) In eArchT+ animals, we found no differences between performance in TI and NI trials taken as groups, both significantly lower than both baseline (p<0.001), and the corresponding trial groups in CTRL animals (TI, p=0.02, NI, p=0.008). In CTRL animals, no differences were found among the three groups (p=0.252 and 0.184). (**C**) Illumination on any given trial resulted in lower performance in the subsequent trial in eArch+, but not CTRL, animals (p=0.015), regardless of whether the illuminated trial is a correct or an incorrect trial. All summary data depicted is mean ± SD.

The online version of this article includes the following source code for figure 3:

**Source code 1.** Performance in a given session, related to *Figure 3*.

**Source code 2.** Probability of correct trial given light delivered, related to *Figures 3 and 4*.

attenuating the silencing effect. As depicted in *Figure 5* (see also *Supplementary file 1G*), the probability of a trial being correct is lower in eArch+, compared to CTRL animals, whenever TI happens in the previous (*Figure 5A* T-1, p=0.01, Bonferroni post hoc test), the second previous (T-2, p=0.033), and the third previous trial (T-3, p=0.036), such difference becoming nonsignificant on condition T-4, when the fourth previous trial was illuminated (T-4, p=0.078). Considering that the inter-trial interval (ITI) is circa 20 s, we can reasonably estimate that some non-illuminated trials occurred between this and the n-previous illuminated one, and that, despite this, the effects of eArch activation are still present, hence after ~3 min. To try and analyze these effects in more detail, without decreasing the sample size to non-interpretable numbers, we quantified performance specifically following sequences of two consecutively non-illuminated trials. We found that after such sequences, there is still a significant impairment in eArch+ animals, in agreement with the previous analysis and with the notion that the effects of eArch silencing outlast its activation by light (*Figure 5B*, p=0.023, Mann-Whitney U).

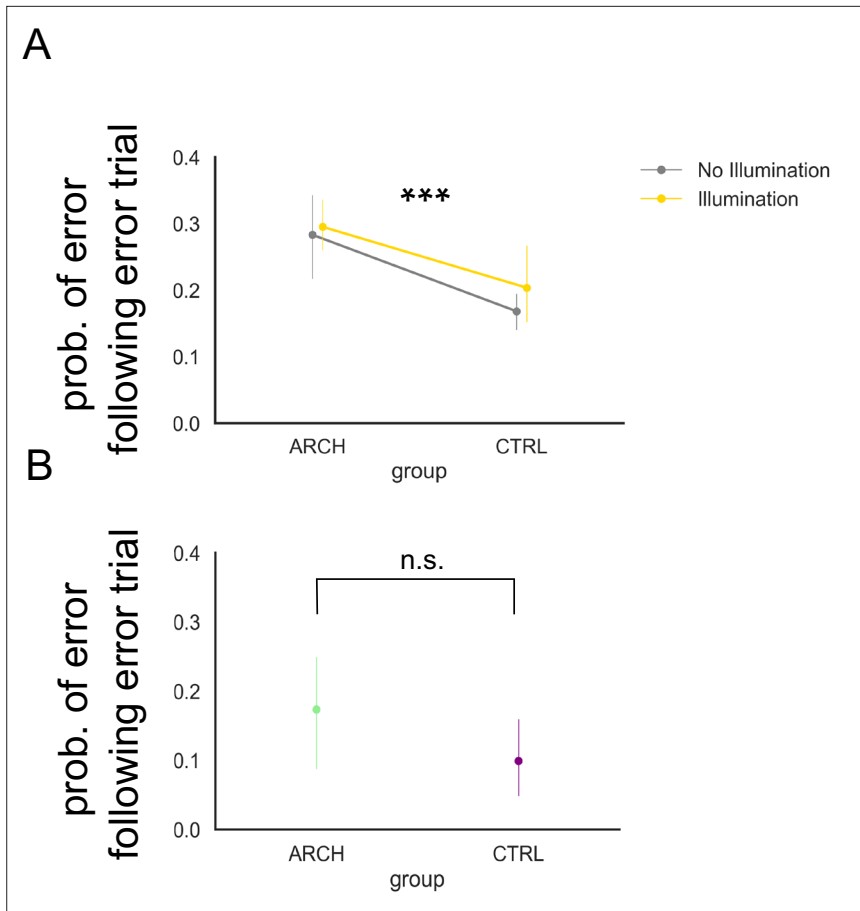

**Figure 4.** Optogenetic silencing of hippocampus (HIPP) terminals in retrosplenial cortex (RSC) causes persistent increase in errors. (**A**) eArch+ animals exhibit persistent errors (p<0.001, Bonferroni post hoc test). (**B**) Persistent errors are absent in baseline (no light delivered) sessions (p=0.172, Bonferroni post hoc test). All summary data depicted is mean ± SD.

## Silencing HIPP terminals in RSC hastens decision processing at the choice point

In previous work, we have examined the role CG neurons might play during spatial WM, by silencing CG and measuring performance in this same DNMP task, and found that silencing CG interferes with performance, with animals spending less time in the decision processes in DNMP trials (*Cruz et al., 2023*). We interpreted this decreased latency as hastiness, with animals taking erroneous decisions based on incomplete cognitive processing. We asked the exact same question in the present work, and quantified latency to the choice point, and latency at the choice point, overall, in error vs correct trials (*Figure 6*). We found that, overall, CTRL animals spend more time at the choice point than eArch+ animals (*Figure 6A*, *Supplementary file 1H*, p=0.02, fixed effect omnibus test). Furthermore, this is true for correct but not error trials both under NI and TI (*Figure 6B*, respectively, p=0.008 and p=0.045, simple effects parameter estimates), suggesting that in CTRL animals, correct decisions involve more time taken at the choice point, contrary to error trials in which no such difference is found.

## Discussion

We expressed eArch3.0 in hippocampal neurons and delivered green light into RSC, a cortical region thought to be involved in the encoding of contextual information relevant to represent future goals and actions, to silence HIPP terminals therein while rats ran a well-known spatial WM task. We found that such manipulation significantly decreased spatial WM performance. Interestingly, we found that

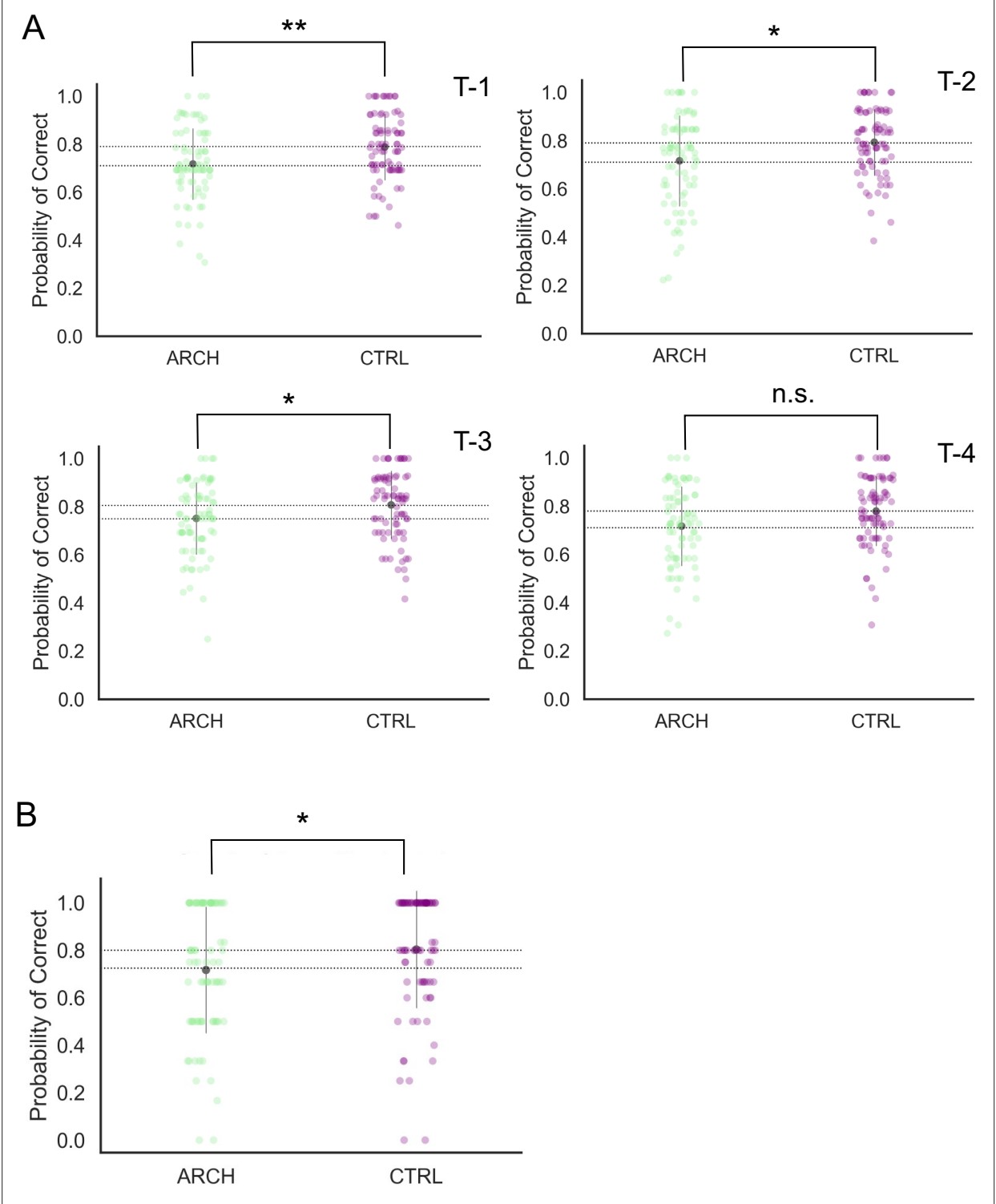

**Figure 5.** Behavioral impairments resulting from optogenetic silencing of hippocampus (HIPP) terminals in retrosplenial cortex (RSC) outlast illumination. (**A**) eArch+ animals exhibit decreased performance once silencing occurred at T-1 (p=0.01, Bonferroni post hoc test), T-2 (p=0.033), T-3 (p=0.036), not T-4 (p=0.078). (**B**) Two consecutively non-illuminated trials after illumination still result in decreased performance on the subsequent trial (p=0.023, Mann-Whitney U). All summary data depicted is mean ± SD.

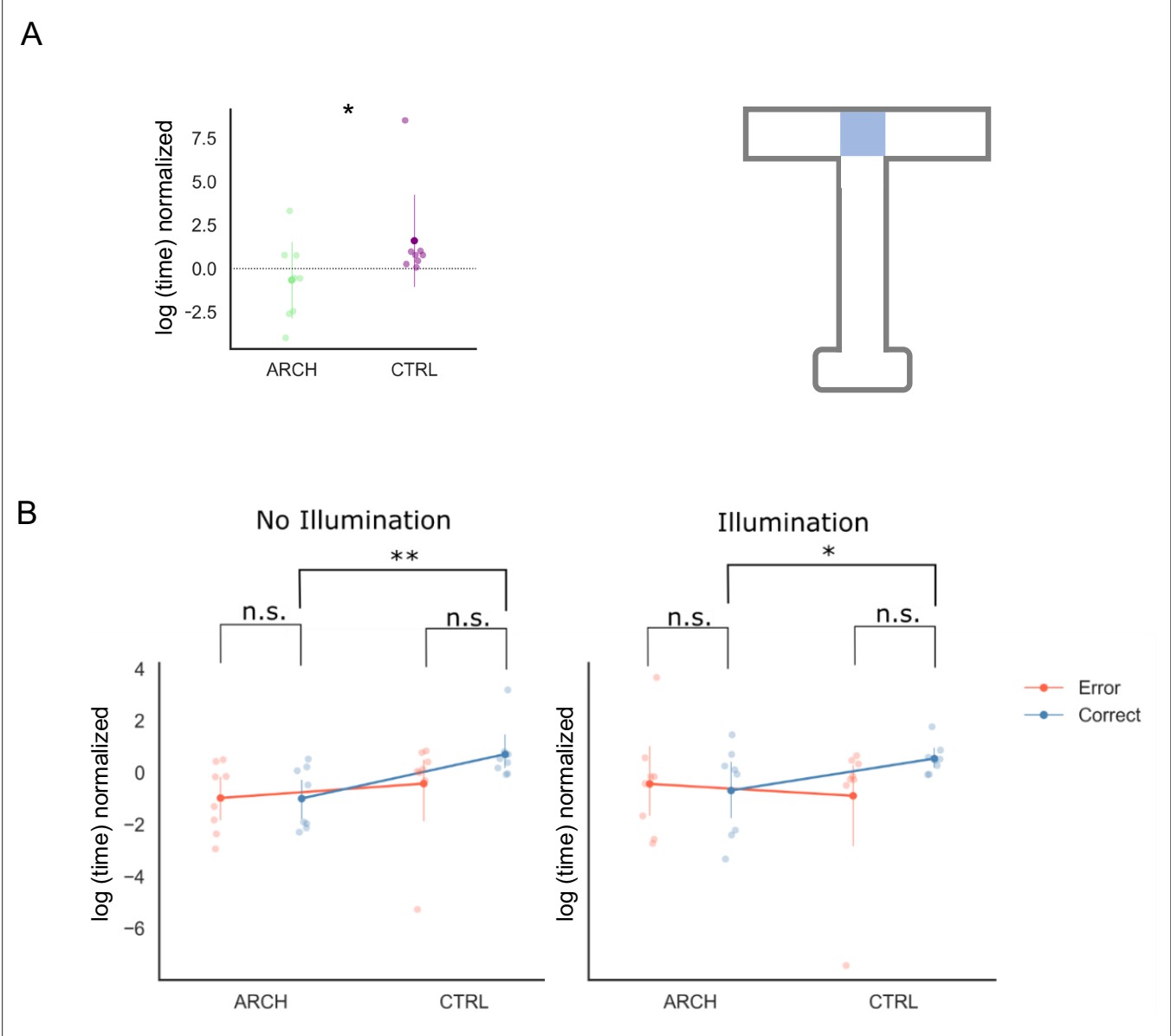

**Figure 6.** Animals spend more time at the choice point in correct trials unless hippocampus (HIPP) terminals in retrosplenial cortex (RSC) are silenced. (**A**) CTRL animals spend more time at the choice point than eArch+ animals (p=0.02, fixed effect omnibus test). (**B**) This is true specifically for correct trials in both NI and TI (respectively, p=0.008 and p=0.045, simple effects parameter estimates). All summary data depicted is mean ± SD.

The online version of this article includes the following source code for figure 6:

**Source code 1.** Time spent at the choice point, related to *Figure 6*.

the effect of hippocampal terminal silencing in RSC during test trials affects interleaved trials during which no light is delivered, thus spreading beyond illumination. We explored this effect further and found that eArch activation lowers performance down to the subsequent three non-illuminated trials, but also that it decreases error-corrective behavior, in which animals use action-outcome information after an error trial (i.e. absence of reward), to correct course in the subsequent trial. Such a prolonged effect finds a precedent in a previous study (*Robinson et al., 2017*) in which silencing entorhinal-HIPP circuits during WM trials results in lower performance in subsequent non-illuminated trials, suggesting that a possible short- to mid-term neural plasticity event disrupts a key process at play during retrieval. Such an event might be related to recent findings showing that eArch activation in CA3-CA1 synaptic responses in vitro hyperpolarizes neurons independent of local membrane voltage (*Krol et al., 2019*;

*Miao et al., 2015*), reducing the amplitude and slope of recorded fEPSP for over 2 min after illumination offset. This effect is achieved by a pH increase that causes a sustained decrease in synaptic efficacy, with lower vesicle release (*El-Gaby et al., 2016*) at synaptic boutons, replicated on another study using striatal-cortical synapses (*Mahn et al., 2016*). We believe such might be the mechanism behind the above-mentioned synaptic plasticity event. These findings imply that the temporal resolution of optogenetic neuron silencing, with neurons effectively unable to generate action potentials, cannot be generalized to the distinct realm of synaptic transmission. Rather, the effects of optogenetics in synaptic physiology might permeate into the synaptic plasticity mechanisms underlying memory acquisition and retrieval.

Finally, as we have seen before with optogenetic silencing of CG neurons during DNMP (*Cruz et al., 2023*), we found that eArch+ animals spend less time at the choice point compared to CTRL. This behavior is in line with hastier, more erroneous, decisions. In CTRL animals specifically, correct trials are characterized by an increased delay at the choice point, as if the correct decision process took place here. This is consistent with previous findings placing the behavioral decision at an equivalent stage in a continuous alternation task, sensitive to RSC permanent lesions, where goal locations are progressively encoded by RSC.

In sum, by reversibly silencing the HIPP synaptic inputs to RSC, we have shown that silencing this input affects spatial WM, possibly via changes in synaptic mechanisms conveying memorized contextual and episodic information about past decisions and action-outcome associations onto medial mesocortical executive decision centers.

## Methods

### Rats

We used 16 (8 eArch and 8 CTRL) male Long-Evans rats aged 4–6 months and weighing 430–600 g (Charles River Laboratories), housed individually and kept under a 12 hr light/dark cycle, gradually deprived to 80–85% of their free-feeding body weight. All procedures were performed in accordance with EU guidelines, approved by the Portuguese National Authority for Animal Health (license number 017548-2021), as well as by the animal well-being office at Instituto de Medicina Molecular João Lobo Antunes (license number AWB_2020_08_MR_NeuralBases).

### Behavioral apparatus and equipment

Experiments were performed on a black acrylic T-maze comprising a central 170 cm long arm, a 25×30 cm$^2$ start platform, and a horizontal arm (88 cm long) extending equally on both sides with two reward ports at their ends (2 cm diameter) where chocolate milk reward was delivered. A black plastic resting box (57×39×42 cm$^3$) was located near the start region to place the rats during the delay and ITI periods. A cable tray above the maze supported the video recording and optogenetic stimulation systems. Both were linked to a computer situated in an adjacent room outside the behavior room. The Bonsai software (*Lopes et al., 2015*) was used to process and control video recording, position tracking, reward delivery, and optogenetic stimulation onset.

To record the behavioral sessions, a Flea 3 PointGrey camera was used, capturing at a rate of 30 fps. The optogenetic stimulation system used was the PlexBright system (Plexon, Dallas, TX, USA), comprising several components. The system consisted of an LD-1 Single Channel LED Driver set at 240 mA, connected to a Dual LED Commutator placed above the maze. This commutator was linked to two green 525 nm LED modules, each attached to a 2-m-long 0.66 NA patch cable with an FC ferrule tip and stainless-steel cladding. This patch tip was connected to an implanted lambda fiber-optic stub (200/230 μm, 0.66 NA) with a 2 mm active length and 5 mm implant length (Optogenix) via a ceramic sleeve. Additionally, a red 5 mm LED was used to track the rats' xy position coordinates. This LED was connected to the tip of the patch cable, close to the fiber-optic stub implants.

### Surgical procedure: viral injections and fiber-optic stub implants

Animals were bilaterally injected with the viral vector AAV5.hSyn-eArch3.0-eYFP and the respective control AAV5.hSyn-eYFP (University of North Carolina at Chapel Hill Vector Core). Viral titers were 4.4×10$^{12}$ vg/ml and 3.3×10$^{12}$ vg/ml, respectively. Before surgery, animals were placed in an induction chamber saturated with 5% isoflurane until absence of the righting reflex. Once the animals were

deeply anesthetized, they were weighed and transferred to a heating pad. An intraperitoneal injection of a ketamine (37.5 mg/kg) and medetomidine (0.5 mg/kg) cocktail (½ dose) was administered, as well as a subcutaneous injection of carprofen (5 mg/kg, ½ dose), and Ringer's lactate every hour to maintain hydration levels. The heads were shaved, lidocaine applied to the ears, and eyes covered with a protection gel (Lubrithal) to prevent drying of the cornea. The animals were installed in the stereotaxic frame secured by ear bars. Anesthesia was maintained with 1–3% isoflurane in oxygen until the end of the procedure. Once animals were placed on the frame, the surgical site was scrubbed with 10% povidone-iodine alternated with 70% ethanol three times before an incision was made to expose the skull. Bregma and lambda were carefully identified and marked on the skull surface. These two landmarks were used to level the skull dorsoventrally. Once the skull was leveled, the craniotomy coordinates were marked on the bone. Before drilling, the bone surface was treated with Baytril to minimize infection.

A high-power microdrill tool was used to drill the location of four anchoring screws on the lateral edges of the parietal bones and to perform three craniotomies: two single-hole craniotomies (−5, ±4), one on each hemisphere, for the microinjection spots; and one craniotomy for the fiber-optic stubs covering both hemispheres. The latter was drilled in an oval shape covering the fiber-optic stub implantation coordinates (−3, +1), (−5, −1). Loose bone pieces were carefully removed with tweezers. To prevent the brain from swelling while the next steps were performed, the craniotomies were covered with hemostatic sponge previously soaked in saline solution.

The viral construct was injected using a Narishige manual microinjection control system operating a glass micropipette plunger, and both held on the stereotaxic frame. The micropipette was filled with mineral oil and placed in the microinjector, after which the viral suspension was aspirated (0.8 µl). Using bregma as a reference, the pipette was carefully positioned over the microinjection coordinates and slowly lowered until reaching the targets: −5, ±4, −2.6 and −5, ±4, −2.8, where 0.4 µl of viral suspension was injected after a 5 min resting period, at a rate of 100 nl/min. Once each injection was done, the micropipette was left in place for 10 min before being slowly retracted. The craniotomies were then covered with hemostatic sponge previously soaked in saline solution and left in place to serve as a barrier between the brain and the dental acrylic (Kerr TAB 2000 fast setting) applied afterward.

In the fiber-optic stub craniotomy, the dura was removed to expose the brain surface over each implant coordinate (−3, +1)(−5, −1). The fiber-optic stubs were secured to the stereotaxic frame, positioned in a 30° angle, and lowered across the two hemispheres to their final target position (−3, +1, −2.86)(−5, −1, −2,86). Following placement of the first fiber-optic stub, it was secured using dental acrylic cement (Kerr TAB 2000 fast setting) connecting it to the anchoring screws, while the opposite hemisphere was protected with hemostatic sponge. The same procedure was followed for the second fiber-optic stub.

Once the dental acrylic was cured, the skin was sutured using absorbable sutures, and 10% povidone-iodine solution was applied around the wound. Carprofen (5 mg/kg, ½ dose) was administered subcutaneously for analgesia. Animals were then placed in a heated cage during the first 24 hr post-surgery for recovery. During this period, ad libitum food, nutritional gel, and water were provided. For the first 7 days, buprenorphine (0.3 mg/kg), a postoperative analgesic, was administered dissolved in chocolate milk.

## Delayed non-matching to place task and protocol

During the week before surgery, rats underwent daily handling and were administered chocolate milk via a 1 ml syringe for habituation. A week post-surgery, the process was repeated, and rats were habituated to the maze. To facilitate this habituation, chocolate milk drops were scattered across the maze and in the reward ports. Once the rats were drinking chocolate milk from the reward ports, they were habituated to do the same with the patch cords attached. Following habituation, the DNMP training sessions started.

Each session consisted of 28 trials, and each trial comprises three epochs: (1) the sample run, (2) the delay, (3) and the test run. (1) In the sample run, one arm was blocked, thereby the rat was forced to visit the only available arm, where a reward awaited; (2) when the rat consumed the reward, it was picked up and placed in the resting box for 15 s. During this delay, the arms and the stem near the choice point were cleaned with 70% ethanol; (3) in the test run, the block from epoch 1 was removed, rats were free to choose which arm to visit (left or right), and were rewarded for choosing the arm

opposite to the sample arm (i.e. non-matching the sample). After the test run, rats were returned to the box for a 20 s ITI and a new trial started.

The sample run arms were balanced (50% left and 50% right) and pseudo-randomized to avoid more than three consecutive runs to the same arm. Rats were scored based on the number of correct and incorrect choices in the test run. Throughout the sessions, the position of the experimenter in the room remained constant. Rats were trained until reaching a performance of at least 75% of correct choices in three consecutive sessions, after which we introduced optogenetic inhibition.

## Optogenetic inhibition protocol

The protocol consisted of 10 consecutive sessions of 28 trials, of which 14 were illuminated in the test epoch (TI) and 14 were not illuminated (NI). This interleaved approach was chosen to prevent compensatory mechanisms. The illuminated trials had the arms pseudo-randomized, ensuring a balanced distribution of left and right sample arms (50% each) and restricted to allow a maximum of three consecutive trials. The onset of illumination was triggered to the position of the animal at 120 cm into the central arm and ended when the rat reached the chosen arm, at 190 cm. Based on the average speed of the animal at this point (~50 cm/s), we estimate illumination time to be ~1.4 s. Each rat completed a total of 280 trials, half of which were subjected to optogenetic inhibition. Upon concluding the experiment, the rats were sacrificed, the implant was extracted, and the fiber-optic stubs were connected to the optogenetic system to verify the emission of light through the optic fiber.

## Histology

At the end of the experiments, animals were sacrificed via an isoflurane overdose followed by an intra-peritoneal injection of sodium pentobarbital (800 mg/kg). Rats were then transcardially perfused with 250 ml of PBS followed by 500 ml of 10% formalin solution. The brains were dissected and kept in a Falcon tube containing 10% formalin solution, covered with aluminum foil, and left at room temperature for 24 hr. Afterward, the brains were transferred to a 15% sucrose solution and kept at 4°C overnight. Once sunk, the brains were embedded in gelatin, frozen in 2-methylbutane liquid nitrogen, and sectioned in 50 µm coronal slices using a cryostat (model CM3050 S, Leica). The fixed brain slices were degelatinized and incubated in DAPI 1:700 (Sigma) for 20 min. These slices were then mounted, coverslipped in Mowiol, and left to dry for 24 hr. The expression of the eArch3.0 viral construct in the HIPP was confirmed using an AxioZoom V16 fluorescence stereo microscope (Zeiss). The expression of the eArch3.0 viral construct in the RSC was confirmed using an LSM 880 confocal point-scanning microscope with Airyscan (Zeiss). CTRL animals underwent the exact same procedure.

## Data and statistical analyses

Datasets were manually assembled containing specific details of each run. Within this dataset, each session's runs were detailed with the following specifications: run number, run type (sample or test), illumination condition (TI or NI), and outcome (correct or incorrect trial). Additionally, the rats' xy positions were tracked via an LED, recorded, and timestamped using Bonsai software (*Lopes et al., 2015*). Custom Python scripts, all deposited in Dryad under the DOI 10.5061/dryad.fxpnvx11z (*Pinto-Correia, 2025*), were used to compute individual runs from these data, ensuring the accuracy of individual runs while excluding any with position data interference. Performance was computed for each experimental session, including baseline sessions, consisting of the three sessions before starting optogenetic inhibition. Performance for illuminated and non-illuminated conditions was calculated as the ratio of correct trials over the total number of illuminated or non-illuminated trials, respectively. Probability computations were conducted for trial accuracy given the previous trial's illumination ($\text{trial}_{n-1}$ to $\text{trial}_{n-4}$) and for error occurrence given the previous error trial as follows:

$$P\,(\text{correct trial}_n \,|\, \text{trial}_{n-x}\,\text{illuminated}) = P\,(\text{trial}_n\,\text{is correct} \cap \text{trial}_{n-x}\,\text{is illuminated}) / P\,(\text{illuminated trials}_{n-x})$$

$$P\,(\text{error trial}_n \,|\, \text{trial}_{n-1}\,\text{error}) = P\,(\text{trial}_n\,\text{is an error} \cap \text{trial}_{n-1}\,\text{is an error}) / P\,(\text{error trials}_{n-1})$$

The time spent at the choice point was determined as the difference between the timestamps of choice point entry and exit. An additional 10 cm was added to the region of interest to account for instances when the rat was stationary but engaged in head poking. Time values were log-transformed to achieve a Gaussian distribution. Normalization to the baseline of log-transformed time in the choice point to the baseline was computed as: [log(latency) − log(baseline mean)]/[log(baseline mean)].

Outliers in time data were identified as values deviating by 2 standard deviations (SD) from the mean and were subsequently removed.

GLMMs were employed to analyze the results. Model selection was based on the Akaike information criterion (AIC), with the chosen model being the one that converged and exhibited the lowest AIC value. The normality of a distribution was assessed using the Shapiro test along with assessment of residuals using quantile-quantile (Q-Q) plots. All the analysis was conducted in Jamovi with the GAMLJ package.

To assess baseline performance differences between eArch and CTRL groups, we used a GLMM with a binomial distribution and logit link function, with individual subjects (i.e. rats) as the random effects variable. The same GLMM, followed by a Bonferroni post hoc test, was conducted to analyze the ratio of correct trials between groups during the experiment and to assess the trial history effect of illumination. The same tests were used to analyze the trial history effect of errors, using as random effect variables the 'session number' and 'run number'. To compare the groups after two consecutive illuminated trials, a Mann-Whitney U test was applied. For analyzing log time in the choice point, a GLMM with a Gaussian distribution and identity link function was applied, considering rat as the random effects variable. To determine significant differences between groups, we used the Wald chi-square test, and for differences between groups and illumination condition, we used simple effects parameter estimates. All code and data are deposited in Dryad under the DOI 10.5061/dryad. fxpnvx11z (*Pinto-Correia, 2025*).

## Lead contact and materials availability

Further information and requests for resources and reagents should be directed to and will be fulfilled by Lead Contact, Miguel Remondes, DVM, PhD (mremondes@medicina.ulisboa.pt). This study did not generate new unique reagents. All data and code are deposited on Dryad under 10.5061/dryad. fxpnvx11z.

## Acknowledgements

We are indebted to IMM's Bioimaging, Comparative Pathology, and Rodent facilities for critical help. We want to thank all the members of the Remondes Lab for fruitful discussions and help. FCT granted a PhD Fellowship (PD/BD/128107/2016, COVID/BD/151601/2021) to BP-C, and an Exploratory Grant (IF/00201/2013), an IMM Director's Fund Award, an Investigator FCT Position at IMM-JLA (IF/00201/2013), a Research Grant (PTDC/MED-NEU/29325/2017), and the 2022.00811.CEECIND Principal Investigator contract to MR.

## Additional information

### Funding

| Funder | Grant reference number | Author |
| --- | --- | --- |
| Fundação para a Ciência e a Tecnologia | PD/BD/128107/2016 | Bárbara Pinto-Correia |
| Fundação para a Ciência e a Tecnologia | IF/00201/2013 | Miguel Remondes |
| Fundação para a Ciência e a Tecnologia | 10.54499/COVID/ BD/151601/2021 | Bárbara Pinto-Correia |
| Fundação para a Ciência e a Tecnologia | PTDC/MED-NEU/29325/2017 | Miguel Remondes |
| Fundação para a Ciência e a Tecnologia | 2022.00811.CEECIND | Miguel Remondes |

The funders had no role in study design, data collection and interpretation, or the decision to submit the work for publication.

## Author contributions

Bárbara Pinto-Correia, Conceptualization, Data curation, Formal analysis, Investigation, Writing - review and editing; Patrícia Caldeira-Bernardo, Formal analysis, Investigation; Miguel Remondes, Conceptualization, Supervision, Funding acquisition, Investigation, Methodology, Writing – original draft, Project administration

## Author ORCIDs

Miguel Remondes ![ORCID] https://orcid.org/0000-0001-9321-3985

## Ethics

All procedures were performed in accordance with EU guidelines, approved by the Portuguese National Authority for Animal Health (license number 017548-2021), as well as by the animal well-being office at Instituto de Medicina Molecular João Lobo Antunes (license number AWB_2020_08_MR_NeuralBases).

Reviewer #2 (Public Review): https://doi.org/10.7554/eLife.96515.3.sa1
Author response https://doi.org/10.7554/eLife.96515.3.sa2

---

# Additional files

## Supplementary files

Supplementary file 1. Generalized linear mixed model tables (A–H).

MDAR checklist

## Data availability

Data is deposited in Dryad. Code is available as a source file in each pertinent figure.

The following dataset was generated:

| Author(s) | Year | Dataset title | Dataset URL | Database and Identifier |
|---|---|---|---|---|
| Pinto-Correia B, Caldeira P, Remondes M | 2025 | Position and time coordinates, performance, and latencies of delayed non-match to place trials performed by rats with silenced hippocampal inputs to RSC | https://doi.org/10.5061/dryad.fxpnvx11z | Dryad Digital Repository, 10.5061/dryad.fxpnvx11z |

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
