## [Editor Report · eLife assessment]

The authors report that optogenetic inhibition of hippocampal axon terminals in retrosplenial cortex impairs the performance of a delayed non-match to place task. Elucidating the role of hippocampal projections to the retrosplenial cortex in memory and decision-making behaviors is **important**. However, the strength of evidence for the paper's claims is **incomplete**.

---

## [Referee Report · Reviewer #2 (Public Review)]

The authors examine the impact of optogenetic inhibition of hippocampal axon terminals in the retrosplenial cortex (RSP) during the performance of a working memory T-maze task. Performance on a delayed non-match-to-place task was impaired by such inhibition. The authors also report that inhibition is associated with faster decision-making and that the effects of inhibition can be observed over several subsequent trials. The work seems reasonably well done and the role of hippocampal projections to retrosplenial cortex in memory and decision-making is very relevant to multiple fields. However, the work should be expanded in several ways before one can make firm conclusions on the role of this projection in memory and behavior.

Comments on revised version:

The authors have provided their comments on the concerns voiced in my first review. I remain of the opinion that the experiments do not extend beyond determining whether disruption of hippocampal to retrosplenial cortex connections impacts spatial working memory. Given the restricted level of inquiry and the very moderate effect of the manipulation on memory, the work, in my opinion, does not provide significant insight into the processes of spatial working memory nor the function of the hippocampal to retrosplenial cortex connection.

---

## [Author Response]

The following is the authors’ response to the original reviews.

**eLife assessment**
The authors report that optogenetic inhibition of hippocampal axon terminals in retrosplenial cortex impairs the performance of a delayed non-match to place task. The significance of findings elucidating the role of hippocampal projections to the retrosplenial cortex in memory and decision-making behaviors is important. However, the strength of evidence for the paper's claims is currently incomplete.
**Public Reviews:**

**Reviewer #1 (Public Review):**
Summary:This is a study on the role of the retrosplenial cortex (RSC) and the hippocampus in working memory. Working memory is a critical cognitive function that allows temporary retention of information for task execution. The RSC, which is functionally and anatomically connected to both primary sensory (especially visual) and higher cognitive areas, plays a key role in integrating spatial-temporal context and in goal-directed behaviors. However, the specific contributions of the RSC and the hippocampus in working memory-guided behaviors are not fully understood due to a lack of studies that experimentally disrupt the connection between these two regions during such behaviors.In this study, researchers employed eArch3.0 to silence hippocampal axon terminals in the RSC, aiming to explore the roles of these brain regions in working memory. Experiments were conducted where animals with silenced hippocampal axon terminals in the RSC performed a delayed non-match to place (DNMP) task. The results indicated that this manipulation impaired memory retrieval, leading to decreased performance and quicker decision-making in the animals. Notably, the authors observed that the effects of this impairment persisted beyond the light-activation period of the opsin, affecting up to three subsequent trials. They suggest that disrupting the hippocampal-RSC connection has a significant and lasting impact on working memory performance.Strengths:They conducted a study exploring the impact of direct hippocampal inputs into the RSC, a region involved in encoding spatial-temporal context and transferring contextual information, on spatial working memory tasks. Utilizing eArch3.0 expressed in hippocampal neurons via the viral vector AAV5-hSyn1-eArch3.0, they aimed to bilaterally silence hippocampal terminals located at the RSC in rats pre-trained in a DNMP task. They discovered that silencing hippocampal terminals in the RSC significantly decreased working memory performance in eArch+ animals, especially during task interleaving sessions (TI) that alternated between trials with and without light delivery. This effect persisted even in non-illuminated trials, indicating a lasting impact beyond the periods of direct manipulation. Additionally, they observed a decreased likelihood of correct responses following TI trials and an increased error rate in eArch+ animals, even after incorrect responses, suggesting an impairment in error-corrective behavior. This contrasted with baseline sessions where no light was delivered, and both eArch+ and control animals showed low error rates.Weaknesses:While I agree with the authors that the role of hippocampal inputs to the RSC in spatial working memory is understudied and merits further investigation, I find that the optogenetic experiment, a core part of this manuscript that includes viral injections, could be improved. The effects were rather subtle, rendering some of the results barely significant and possibly too weak to support major conclusions.

We thank Reviewer#1 for carefully and critically reading our manuscript, and for the valuable comments provided. The judged “subtlety” of the effects stems from a perspective according to which a quantitatively lower effect bears less biological significance for cognition. We disagree with this perspective and find it rather reductive for several reasons.

Once seen in the context of the animal’s ecology, subtle impairments can be life-threatening precisely because of their subtlety, leading the animal to confidently rely on a defective capacity, for such events as remembering the habitual location of a predator, or food source.

Also, studies in animal cognition often undertake complete, rather than graded, suppression of a given mechanism (in the same sense as that of “knocking out” a gene that is relevant for behaviour), leading to a gravelly, rather that gradually, impaired model system, to the point of not allowing a hypothetical causal link to be mechanistically revealed beyond its mere presence. This often hinders a thorough interpretation of the perturbed factor’s role. If a caricatural analogy is allowed, it would be as if we were to study the role of an animal’s legs by chopping them both off and observing the resulting behaviour.

In our study we conclude that silencing HIPP inputs in RSC perturbs cognition enough to impair behaviour while not disabling the animal entirely, as such allowing for behaviour to proceed, and for our observation of graded, decreased (not absent), proficiency under optogenetic silencing. So rather than weak, we would say the results are statistically significant, and biologically realistic.

Additionally, no mechanistic investigation was conducted beyond referencing previous reports to interpret the core behavioral phenotypes.

We fully agree with this being a weakness, as we wish we could have done more mechanistic studies to find out exactly what is Arch activation doing to HIPP-RSC transmission, which neurons are being affected, and perhaps in the future dissect its circuit determinants. We have all these goals very present and hope we can address them soon.

**Reviewer #2 (Public Review):**
The authors examine the impact of optogenetic inhibition of hippocampal axon terminals in the retrosplenial cortex (RSP) during the performance of a working memory T-maze task. Performance on a delayed non-match-to-place task was impaired by such inhibition. The authors also report that inhibition is associated with faster decision-making and that the effects of inhibition can be observed over several subsequent trials. The work seems reasonably well done and the role of hippocampal projections to retrosplenial cortex in memory and decision-making is very relevant to multiple fields. However, the work should be expanded in several ways before one can make firm conclusions on the role of this projection in memory and behavior.

We thank Reviewer#2 for carefully and critically reading our manuscript, and for the valuable comments provided.

(1) The work is very singular in its message and the experimentation. Further, the impact of the inhibition on behaviour is very moderate. In this sense, the results do not support the conclusion that the hippocampal projection to retrosplenial cortex is key to working memory in a navigational setting.

As we have mentioned in response to Reviewer#1, the judged “very moderate” effect stems from a perspective according to which a quantitatively lower effect bears less biological significance for cognition, precluding its consideration as “key” for behaviour. We disagree with this perspective and find it rather reductive for several reasons. Once seen in the context of the animal’s ecology, quantitatively lower impairments in working memory are no less key for this cognitive capacity, and can be life-threatening precisely because of their subtlety, leading the animal to confidently rely on a defective capacity, for such events as remembering the habitual location of a predator, or food source. Furthermore, studies in animal cognition often undertake complete, rather than graded, suppression of a given mechanism (in the same sense as “knocking out” a gene that is relevant for behaviour), leading to a gravelly, rather that gradually, impaired model system, to the point of not allowing a hypothetical causal link to be mechanistically revealed beyond its mere presence. This often hinders a thorough interpretation of its role.

In our study we conclude that silencing HIPP inputs in RSC perturbs behaviour enough to impair behaviour while not disabling the animal entirely, as such allowing for behaviour to proceed, and our observation of graded, decreased (not absent), proficiency under optogenetic silencing. So rather than weak, we would say the results are statistically significant, and biologically realistic.

(2) There are no experiments examining other types of behavior or working memory. Given that the animals used in the studies could be put through a large number of different tasks, this is surprising. There is no control navigational task. There is no working memory test that is non-spatial. Such results should be presented in order to put the main finding in context.

It is hard to gainsay this point. The more thorough and complete a behavioural characterization is, the more informative is the study, from every angle you look at it. While we agree that other forms of WM would be quite interesting in this context, we also cannot ignore the fact that DNMP is widely tested as a WM task, one that is biologically plausible, sensitive to perturbations of neural circuitry know to be at play therein, and fully accepted in the field. Faced with the impossibility of running further studies, for lack of additional funding and human resources, we chose to run this task.

A control navigational task would, in our understanding, be used to assess whether silencing HIPP projections to RSC would affect (spatial?) navigation, rather than WM, thus explaining the observed impairment. To this we have the following to say: Spatial Navigation is a very basic cognitive function, one that relies on body orientation relative to spatial context, on keeping an updated representation of such spatial context, (“alas”, as memory), and on guiding behaviour according to acquired knowledge about spatial context. Some of these functions are integral to spatial working memory, as such, they might indeed be affected.

Dissecting the determinants of spatial WM is indeed an ongoing effort, one that was not the intention of the current study, but also one that we have very present, in hope we can address in the future.

A non-spatial WM task would indeed vastly solidify our claims beyond spatial WM, onto WM. We have, for this reason, changed the title of the manuscript which now reads “spatial working memory”.

(3) The actual impact of the inhibition on activity in RSP is not provided. While this may not be strictly necessary, it is relevant that the hippocampal projection to RSP includes, and is perhaps dominated by inhibitory inputs. I wonder why the authors chose to manipulate hippocampal inputs to RSP when the subiculum stands as a much stronger source of afferents to RSP and has been shown to exhibit spatial and directional tuning of activity. The points here are that we cannot be sure what the manipulation is really accomplishing in terms of inhibiting RSP activity (perhaps this explains the moderate impact on behavior) and that the effect of inhibiting hippocampal inputs is not an effective means by which to study how RSP is responsive to inputs that reflect environmental locations.

We fully agree that neural recordings addressing the effect of silencing on RSC neural activity is relevant. We do wish we could have provided more mechanistic studies, to find out exactly what is Arch activation doing to HIPP-RSC transmission, which neurons are being affected, and thus dissecting its circuit determinants. We have all these goals very present and hope we can address them soon. Subiculum, which we mention in the Introduction, is indeed a key player in this complex circuitry, one whose hypothetical influence is the subject of experimental studies which will certainly reveal many other key elements.

(4) The impact of inhibition on trials subsequent to the trial during which optical stimulation was actually supplied seems trivial. The authors themselves point to evidence that activation of the hyperpolarizing proton pump is rather long-lasting in its action. Further, each sample-test trial pairing is independent of the prior or subsequent trials. This finding is presented as a major finding of the work, but would normally be relegated to supplemental data as an expected outcome given the dynamics of the pump when activated.

We disagree that this finding is “trivial”, and object to the considerations of “normalcy”, which we are left wondering about.

In lack of neurophysiological experiments (for the reasons stated above) to address this interesting finding, we chose to interpret it in light of (the few) published observations, such being the logical course of action in scientific reporting, given the present circumstances.

Evidence for such a prolonged effect in the context of behaviour is scarce (to our knowledge only the one we cite in the manuscript). As such, it is highly relevant to report it, and give it the relevance we do in our manuscript, rather than “relegating it to supplementary data”, as the reviewer considers being “normal”.

In the DNMP task the consecutive sample-test pairs are explicitly not independent, as they are part of the same behavioural session. This is illustrated by the simple phenomenon of learning, namely the intra-session learning curves, and the well-known behavioral trial-history effects. The brain does not simply erase such information during the ITI.

(5) In the middle of the first paragraph of the discussion, the authors make reference to work showing RSP responses to "contextual information in egocentric and allocentric reference frames". The citations here are clearly deficient. How is the Nitzan 2020 paper at all relevant here?

Nitzan 2020 reports the propagation of information from HIPP to CTX via SUB and RSC, thus providing a conduit for mnemonic information between the two structures, alternative to the one we target, thus providing thorough information concerning the HIPP-RSC circuitry at play during behaviour.

Alexander and Nitz 2015 precisely cite the encoding, and conjunction, of two types of contextual information, internal (ego-) and external (allocentric).

The subsequent reference is indeed superfluous here.

We thank the Reviewer#2 for calling our attention to the fact that references for this information are inadequate and lacking. We have now cited (Gill et al., 2011; Miller et al., 2019; Vedder et al., 2017) and refer readers to the review (Alexander et al., 2023) for the purpose of illustrating the encoding of information in the two reference frames. In addition, we have substantially edited the Introduction and Discussion sections, and suppressed unnecessary passages.

(6) The manuscript is deficient in referencing and discussing data from the Smith laboratory that is similar. The discussion reads mainly like a repeat of the results section.

Please see above. We thank Reviewer#2 for this comment, we have now re-written the Discussion such that it is less of a summary of the Results and more focused on their implications and future directions.

**Response to recommendations for the authors:**

**Reviewer #1 (Recommendations For The Authors):**
MajorLine 101: Even with the tapered lambda fibre optic stub, if the fibre optics were longitudinally staggered by 2 millimetres, they would deliver light to diagonal regions in the horizontal plane rather than covering the full length of the RSC. Is this staggering pattern randomized or fixed? Additionally, Figure 1C is a bit misleading, as the light distribution pattern from the tapered fibre optic is likely to be more concentrated near the surface of the fibre, rather than spreading widely in a large spherical pattern.

The staggering is fixed. The elliptical (not spherical) contour in Fig 1C is not meant to convey any quantitative information, but rather to visually orient the reader towards the directions into which light will likely propagate, the effects of which we do not attempt to estimate here. We have made this contour smaller.

Line 119: The authors demonstrate the viral expression pattern of a representative animal and the overall expression patterns of all other animals in Figure 1 and the Supplementary Figures. However, numerous cases in the Supplementary Figures exhibit viral leakages and strong expressions in adjacent cortical and thalamic areas. Although there is a magnified view of the RSC's expression pattern in Figure 1, authors should show the same way in the supplemental data as well. Additionally, the degree of viral expression in the hippocampal subregions varies substantially across animals. This variation is concerning and impacts the interpretation of the results.

The viral construct was injected in the HIPP at coordinates based on our previous work (Ferreira-Fernandes et al., 2019) wherein injections of a similar vector in mid-dorsal HIPP resulted in widespread expression throughout the medial mesocortex AP extent, RSC through CG, as well as other areas in which HIPP establishes synapses. These were studied in detail then, by estimating the density of axon terminals. In the present work we did not acquire high-mag images of all slices, since they were too expensive, and we had this information from the study above. Still, we have now added further examples of high-mag images taken from eArch and CTRL animals.

We believe it is important here to mention the fact that the virus we use, AAV5, only travels anterograde and is static (i.e. it does not travel transynaptically).

Variations in viral expression are to be expected even if injections happen in the exact same way. It is crucial then, that fibre positioning is constant across animals, to guarantee that its relationship with viral expression is thence consistent, and to render irrelevant whatever off-target expression of the viral construct. We have ascertained this condition post-mortem in all our animals.

Line 124: Another point regarding the viral expressions and optical fibre implants used to inhibit the HIPP-RSC pathway is that the RSC and HIPP extend substantially along the anterior-posterior axis. The authors should demonstrate how the viral expression is distributed along this axis and indicate where the tip of the tapered optical fibre ended by marking it in the histological images. This information is crucial to confirm the authors' claim that the hippocampal projection terminals were indeed modulated by optical light. Also, the manuscript would benefit from details about the power/duration and/or modulation of the light used.

In both Figures 1 and S1 panels we can clearly see the tracks formed by the fibres. This provides examples of such dual angle placement vis a vis the expression of the construct, demonstrating that the former is fully targeted towards the latter. We have added markers to highlight these tracks and an example of a “full” track in figure S1. We did not have animals deviating from this relative positioning to any significant extent. The methods section mentions illumination power as 240mA, and we have now added estimated illumination time as well.

Line 141: The authors should include data on task performance during learning and baseline sessions for each animal, to demonstrate that they fully grasped the task rules and that achieving a 75% performance ratio was sufficient.

DNMP is a standard WM task used for many decades, in which animals reach performances above 75% in 4-8 sessions. We have used it extensively, and never saw any deviations from this learning rate and curve. We ran daily sessions until animals reached 75%, and thereafter until they maintained this performance, or above, for three consecutive sessions (the data points we show). We saw no deviations from what is published, nor from what is our own extensive experience, and thence are fully confident that all animals included in this manuscript grasped task rules.

Line 146: While the study focused on inhibiting inputs during the test run (retrieval phase), it would be beneficial to also inhibit inputs during the sample run (encoding phase) and the delay period. This would help confirm whether the silencing affects only working memory retrieval, or if it also impacts encoding and maintenance.

We agree, it would be very interesting to determine if there are any effects of silencing HIPP RSC terminals during Sample. However, since there is a limit to the number of trials per session, and to the total number of sessions, we could not run the three manipulations within each session of our experimental design, as that would lower the number of trials per condition to an extent that would affect statistical power. Silencing HIPP RSC terminals during Sample would best be a separate experiment, asking a different question, and perhaps within an experimental design distinct from the one envisioned.

A very important point here relates to the fact that the effects of optogenetic manipulation do not limit themselves to the illumination epoch, in fact they extend far beyond onto the 3rd trial post-illumination. The insertion of Sample-illuminated trials interleaved in the same session would fundamentally affect the interpretation of experimental results, as we could not attribute lower performances to the effects in either or both manipulated epochs.

Line 225: Figure 5 illustrates that silencing the inputs results in an extended impairment of working memory performance. However, it's unclear if there are any behavioural changes during the sample run. The inhibition could potentially affect encoding in the subsequent sample run, considering the inter-trial interval (ITI) is only 20 seconds.

From the observation of behaviour and the analysis of our data, we saw no overt “behavioural changes during the sample run”, as latencies and speeds were essentially unchanged.

If what is meant by your comment is the effect of optogenetic manipulation being protracted from the Test towards the Sample epoch, we find this unlikely. Conservatively, we estimate the peak of our optogenetic manipulation to occur around the time light is delivered, the Test phase, rather than 20-30 secs later.

In theory, any effect of optogenetic silencing of HIPP terminals in RSC can cause disturbances in encoding or Sample, the ITI itself, and the epoch in which mnemonic information retrieved from the Sample epoch is confronted with the contextual information present during Test, leading to a decision. This is regardless of the illumination epoch, and even if the effect of optogenetic manipulation is not prolonged in time.

Since in our experiments we specifically target the Test epoch, and there is, in all likelihood, a decaying magnitude of neurophysiological effects, manifest in the reported decaying nature of the manipulation mechanism, and in our observed decrease of behavioural proficiency from subsequent trials 1:4, we are convinced that a conservative interpretation is that our major effect is concentrated in the epoch in which we deliver light - the Test epoch, the consequences of which (possibly related to short term plasticity events taking place within the HIPP-RSC neural circuit) extending further in time.

Line 410: The methods section on the surgical procedure could be clearer, particularly regarding the coordinates for microinjection and fibre implantation. A more precise description would aid reader comprehension.

The now-reported injection and implantation coordinates include the numbers corresponding to the distances, in mm, from Bregma to the targets, in the three stereotaxic dimensions considered: antero-posterior, medial-lateral left and right, and dorso-ventral, as well as the angle at which the fibres were positioned. We have added labels to the figures to highlight the fibreoptic track locations. We will be happy to provide further details as deemed necessary.

Line 461: It would be helpful to know if each animal displayed a preference for the left or right side. Including a description or figure showing that the performance ratio exceeded 75% in both left and right trials would provide a more comprehensive understanding of the animals' behaviour.

In the DNMP, an extensively used and documented WM task, it is an absolute pre-condition that no animals are biased to either side. As such, we did not use any animal that showed such bias.

We have not observed this to be the case in any of our candidate animals, nor would we use any animal exhibiting such a preference.

MinorLine 25: In the INTRODUCTION section, the authors introduce ego-centric and allocentric variables in the RSC. However, if they intend to discuss this feature, there is no supporting data for ego-centric or allocentric variables in the Results section.

We agree. The extent of the discussion of ego vs allo-centric variables in our manuscript might venture a bit out of the main subject. It was included to provide wider context to our reporting of the data, considering that spatial working memory is indeed one instance in which egocentric- and allocentric-referenced cognitive mechanisms confront each other, and one in which silencing the HIPP input to a cortical region thence involved would likely disturb ensuing computations. We have now substantially edited the manuscript’s Introduction and Discussion, sections, namely toning down this aspect.

Line 125: In the section title, DNMT -> DNMP obviously.

We have corrected this passage.

Figures: The quality of the figure panels does not meet the expected standards. For example, scale bars are missing in many panels (e.g., Figure 1A bottom, 1B, 1C, S1), figure labels are misaligned (as seen in Figure 3A-B compared to 3C, same with Figure 5), and there is inconsistency in color schemes (e.g., Figure 3C versus Figure 6, where 'Error' versus 'Correct' is depicted using green versus blue, respectively).

We have now corrected these inconsistencies and mistakes.